# How many classifiers do we need?

**Hyunsuk Kim**
Department of Statistics
University of California, Berkeley
hyskim7@berkeley.edu

**Liam Hodgkinson**
School of Mathematics and Statistics
University of Melbourne, Australia
lhodgkinson@unimelb.edu.au

**Ryan Theisen**
Harmonic Discovery
ryan@harmonicdiscovery.com

**Michael W. Mahoney**
ICSI, LBNL, and Dept. of Statistics
University of California, Berkeley
mmahoney@stat.berkeley.edu

## Abstract

As performance gains through scaling data and/or model size experience diminishing returns, it is becoming increasingly popular to turn to ensembling, where the predictions of multiple models are combined to improve accuracy. In this paper, we provide a detailed analysis of how the disagreement and the polarization (a notion we introduce and define in this paper) among classifiers relate to the performance gain achieved by aggregating individual classifiers, for majority vote strategies in classification tasks. We address these questions in the following ways. (1) An upper bound for polarization is derived, and we propose what we call a neural polarization law: most interpolating neural network models are 4/3-polarized. Our empirical results not only support this conjecture but also show that polarization is nearly constant for a dataset, regardless of hyperparameters or architectures of classifiers. (2) The error rate of the majority vote classifier is considered under restricted entropy conditions, and we present a tight upper bound that indicates that the disagreement is linearly correlated with the error rate, and that the slope is linear in the polarization. (3) We prove results for the asymptotic behavior of the disagreement in terms of the number of classifiers, which we show can help in predicting the performance for a larger number of classifiers from that of a smaller number. Our theoretical findings are supported by empirical results on several image classification tasks with various types of neural networks.

## 1 Introduction

As performance gains through scaling data and/or model size experience diminishing returns, it is becoming increasingly popular to turn to *ensembling*, where the predictions of multiple models are combined, both to improve accuracy and to form more robust conclusions than any individual model alone can provide. In some cases, ensembling can produce substantial benefits, particularly when increasing model size becomes prohibitive. In particular, for large neural network models, *deep ensembles* [LPB17] are especially popular. These ensembles consist of independently trained models on the same dataset, often using the same hyperparameters, but starting from different initializations.

The cost of producing new classifiers can be steep, and it is often unclear whether the additional performance gains are worth the cost. Assuming that constructing two or three classifiers is relatively cheap, procedures capable of deciding whether to continue producing more classifiers are needed. To do so requires a precise understanding of how to predict ensemble performance. Of particular interest are majority vote strategies in classification tasks, noting that regression tasks can also be formulated in this way by clustering outputs. In this case, one of the most effective avenues for

38th Conference on Neural Information Processing Systems (NeurIPS 2024).

predicting performance is the *disagreement* [JNBK22, BJRK22]: measuring the degree to which classifiers provide different conclusions over a given dataset. Disagreement is concrete, easy to compute, and strongly linearly correlated with majority vote prediction accuracy, leading to its use in many applications. However, *a priori*, the precise linear relationship between disagreement and accuracy is unclear, preventing the use of disagreement for predicting ensemble performance.

Our goal in this paper is to go beyond disagreement-based analysis to provide a more quantitative understanding of the number of classifiers one should use to achieve a desired level of performance in modern practical applications, in particular for neural network models. In more detail, our contributions are as follows.

(i) We introduce and define the concept of **polarization**, a notion that measures the higher-order dispersity of the error rates at each data point, and which indicates how polarized the ensemble is relative to the ground truth. We state and prove an upper bound for polarization (Theorem 1). Inspired by the theorem, we propose what we call a **neural polarization law** (Conjecture 1): most interpolating (Definition 2) neural network models are 4/3-polarized. We provide empirical results supporting the conjecture (Figures 1 and 2).

(ii) Using the notion of polarization, we develop a refined set of **bounds on the majority vote test error rate**. For one, we provide a sharpened bound for any ensembles with a **finite number** of classifiers (Corollary 1). For the other, we offer a new, tighter bound under an additional condition on the **entropy** of the ensemble (Theorem 4). We provide empirical results that demonstrate our new bounds perform significantly better than the existing bounds on the majority vote test error (Figure 3).

(iii) The **asymptotic behavior of the majority vote error rate** is determined as the number of classifiers increases (Theorem 5). Consequently, we show that we can predict the performance for a larger number of classifiers from that of a smaller number. We provide empirical results that show such predictions are accurate across various pairs of model architecture and dataset (Figure 4).

In Section 2, we define the notations that will be used throughout the paper, and we introduce upper bounds for the error rate of the majority vote from previous work. The next three sections are the main part of the paper. In Section 3, we introduce the notion of polarization, $\eta_\rho$, which plays a fundamental role in relating the majority vote error rate to average error rate and disagreement. We explore the properties of the polarization and present empirical results that corroborate our claims. In Section 4, we present tight upper bounds for the error rate of the majority vote for ensembles that satisfy certain conditions; and in Section 5, we prove how disagreement behaves in terms of the number of classifiers. All of these ingredients are put together to estimate the error rate of the majority vote for a large number of classifiers using information from only three sampled classifiers. In Section 6, we provide a brief discussion and conclusion. Additional material is presented in the appendices.

## 2 Preliminaries

In this section, we introduce notation that we use throughout the paper, and we summarise previous work on the performance of the majority vote error rate.

### 2.1 Notations

We focus on $K$-class classification problems, with features $X \in \mathcal{X}$, labels $Y \in [K] = \{1, 2, ..., K\}$ and feature-label pairs $(X, Y) \sim \mathcal{D}$. A classifier $h : \mathcal{X} \to [K]$ is a function that maps a feature to a label. We define the error rate of a single classifier $h$, and the disagreement and the tandem loss [MLIS20] between two classifiers, $h$ and $h'$, as the following:

$$\text{Error rate} : L(h) = \mathbb{E}_{\mathcal{D}}[\mathbb{1}(h(X) \neq Y)]$$
$$\text{Disagreement} : D(h, h') = \mathbb{E}_{\mathcal{D}}[\mathbb{1}(h(X) \neq h'(X))]$$
$$\text{Tandem loss} : L(h, h') = \mathbb{E}_{\mathcal{D}}[\mathbb{1}(h(X) \neq Y)\mathbb{1}(h'(X) \neq Y)],$$

where the expectation $\mathbb{E}_{\mathcal{D}}$ is used to denote $\mathbb{E}_{(X,Y)\sim\mathcal{D}}$. Next, we consider a distribution of classifiers, $\rho$, which may be viewed as an *ensemble* of classifiers. This distribution can represent a variety of different cases. Examples include: (1) a discrete distribution over finite number of $h_i$, e.g., a weighted sum of $h_i$; and (2) a distribution over a parametric family $h_\theta$, e.g., a distribution of classifiers resulting

from one or multiple trained neural networks. Given the ensemble $\rho$, the *(weighted) majority vote* $h_\rho^{\mathrm{MV}} : \mathcal{X} \to [K]$ is defined as

$$h_\rho^{\mathrm{MV}}(x) = \arg\max_{y \in [K]} \mathbb{E}_\rho[\mathbb{1}(h(x) = y)].$$

Again, $\mathbb{E}_\rho$ denotes $\mathbb{E}_{h \sim \rho}$, and we use $\mathbb{E}_\rho, \mathbb{E}_{\rho^2}, \mathbb{P}_\rho$ for $\mathbb{E}_{h \sim \rho}, \mathbb{E}_{(h,h') \sim \rho^2}, \mathbb{P}_{h \sim \rho}$, respectively, throughout the paper. In this sense, $\mathbb{E}_\rho[L(h)]$ represents *the average error rate* under a distribution of classifiers $\rho$ and $\mathbb{E}_{\rho^2}[D(h, h')]$ represents *the average disagreement* between classifiers under $\rho$. Hereafter, we refer to $\mathbb{E}_\rho[L(h)], \mathbb{E}_{\rho^2}[D(h, h')]$, and $L(h_\rho^{\mathrm{MV}})$ as the **average error rate**, the **disagreement**, and the **majority vote error rate**, respectively, with

$$L(h_\rho^{\mathrm{MV}}) = \mathbb{E}_\mathcal{D}[\mathbb{1}(h_\rho^{\mathrm{MV}}(X) \neq Y)].$$

Lastly, we define the point-wise error rate, $W_\rho(X, Y)$, which will serve a very important role in this paper (for clarity, we will denote $W_\rho(X, Y)$ by $W_\rho$ unless otherwise necessary):

$$W_\rho(X, Y) = \mathbb{E}_\rho[\mathbb{1}(h(X) \neq Y)]. \tag{1}$$

## 2.2 Bounds on the majority vote error rate

The simplest relationship between the majority vote error $L(h_\rho^{\mathrm{MV}})$ and the average error rate $\mathbb{E}_\rho[L(h)]$ was introduced in [McA98]. It states that the error in the majority vote classifier cannot exceed twice the average error rate:

$$L(h_\rho^{\mathrm{MV}}) \leq 2\mathbb{E}_\rho[L(h)] \tag{2}$$

A simple proof for this relationship can be found in [MLIS20] using Markov's inequality. Although (2) does not provide useful information in practice, it is worth noting that this bound is, in fact, tight. There exist pathological examples where $h_\rho^{\mathrm{MV}}$ exhibits twice the average error rate (see Appendix C in [TKY$^+$24]). This suggests that we can hardly obtain a useful or tighter bound by relying on only the "first-order" term, $\mathbb{E}_\rho[L(h)]$.

Accordingly, more recent work constructed bounds in terms of "second-order" quantities, $\mathbb{E}_{\rho^2}[L(h, h')]$ and $\mathbb{E}_{\rho^2}[D(h, h')]$. In particular, [LMRR17] and [MLIS20] designed a so-called *C-bound* using the Chebyshev-Cantelli inequality, establishing that, if $\mathbb{E}_\rho[L(h)] < 1/2$, then

$$L(h_\rho^{\mathrm{MV}}) \leq \frac{\mathbb{E}_{\rho^2}[L(h, h')] - \mathbb{E}_\rho[L(h)]^2}{\mathbb{E}_{\rho^2}[L(h, h')] - \mathbb{E}_\rho[L(h)] + \frac{1}{4}}. \tag{3}$$

As an alternative approach, [MLIS20] incorporated the disagreement $\mathbb{E}_{\rho^2}[D(h, h')]$ into the bound as well, albeit restricted to the binary classification problem, to obtain:

$$L(h_\rho^{\mathrm{MV}}) \leq 4\mathbb{E}_\rho[L(h)] - 2\mathbb{E}_{\rho^2}[D(h, h')]. \tag{4}$$

While (3) and (4) may be tighter in some cases, once again, there do exist pathological examples where this bound is as uninformative as the first-order bound (2). Motivated by these weak results, [TKY$^+$24] take a new approach by restricting $\rho$ to be a "good ensemble," and introducing the *competence* condition (see Definition 3 in our Appendix A). Informally, competent ensembles are those where it is more likely—in average across the data—that more classifiers are correct than not. Based on this notion, [TKY$^+$24] prove that competent ensembles are guaranteed to have weighted majority vote error *smaller* than the weighted average error of individual classifiers:

$$L(h_\rho^{\mathrm{MV}}) \leq \mathbb{E}_\rho[L(h)]. \tag{5}$$

That is, the majority vote classifier is always beneficial. Moreover, [TKY$^+$24] proves that any competent ensemble $\rho$ of $K$-class classifiers satisfy the following inequality.

$$L(h_\rho^{\mathrm{MV}}) \leq \frac{4(K-1)}{K} \left( \mathbb{E}_\rho[L(h)] - \frac{1}{2}\mathbb{E}_{\rho^2}[D(h, h')] \right). \tag{6}$$

We defer further discussion of competence to Appendix A, where we introduce simple cases for which competence does *not* hold. In these cases, we show how one can overcome this issue so that the bounds (5) and (6) still hold. In particular, in Appendix A.3, we provide an example to show the bound (6) is tight.

# 3 The Polarization of an Ensemble

In this section, we introduce a new quantity, $\eta_\rho$, which we refer to as the *polarization* of an ensemble $\rho$. First, we provide examples as to what this quantity represents and draw a connection to previous studies. Then, we present theoretical and empirical results that show this quantity plays a fundamental role in relating the majority vote error rate to average error rate and disagreement. In Theorem 1, we prove an upper bound for the polarization $\eta_\rho$, which highlights a fundamental relationship between the polarization and the constant $\frac{4}{3}$. Inspired from the theorem, we propose Conjecture 1 which we call a *neural polarization law*. Figures 1 and 2 present empirical results on an image recognition task that corroborates the conjecture.

We start by defining the polarization of an ensemble. In essence, the polarization is an improved (smaller) coefficient on the Markov's inequality on $\mathbb{P}_\mathcal{D}(W_\rho > 0.5)$, where $W_\rho$ is the point-wise error rate defined as equation (1). It measures how much the ensemble is "polarized" from the truth, with consideration of the distribution of $W_\rho$.

**Definition 1** (POLARIZATION). *An ensemble $\rho$ is $\eta$-polarized if*

$$\eta \, \mathbb{E}_\mathcal{D}[W_\rho^2] \geq \mathbb{P}_\mathcal{D}(W_\rho > 1/2). \tag{7}$$

*The **polarization** of an ensemble $\rho$ is*

$$\eta_\rho := \frac{\mathbb{P}_\mathcal{D}(W_\rho > 1/2)}{\mathbb{E}_\mathcal{D}[W_\rho^2]}, \tag{8}$$

*which is the smallest value of $\eta$ satisfies inequality (7).*

Note that the polarization always takes a value in $[0, 4]$, due to the positivity constraint and Markov's inequality. Also note that ensemble $\rho$ with polarization $\eta_\rho$ is $\eta$-polarized for any $\eta \geq \eta_\rho$.

To understand better what this quantity represents, consider the following examples. The first example demonstrates that polarization increases as the majority vote becomes more polarized from the truth, while the second example demonstrates how polarization increases when the constituent classifiers are more evenly split.

**Example 1.** Consider an ensemble $\rho$ where 75% of classifiers output Label 1 with probability one, and the other 25% classifiers output Label 2 with probability one.

- **Case 1.** *The true label is Label 1 for the whole data.*
  In this case, the majority vote in $\rho$ results in zero error rate. The point-wise error rate $W_\rho$ is 0.25 on the entire dataset, and thus $\mathbb{P}_\mathcal{D}(W_\rho > 0.5) = 0$. The polarization $\eta_\rho$ is 0.

- **Case 2.** *The true label is Label 1 for half of the data and is Label 2 for the other half.*
  In this case, the majority vote is only correct for half of the data. The point-wise error rate $W_\rho$ is 0.25 for this half, and is 0.75 for the other half. The polarization $\eta_\rho$ is $0.5/0.3125 = 1.6$.

- **Case 3.** *The true label is Label 2 for the whole data.*
  In this case, the majority vote in $\rho$ is wrong on every data point. The point-wise error rate $W_\rho$ is 0.75 on the entire dataset and thus $\mathbb{P}_\mathcal{D}(W_\rho > 0.5) = 1$. The polarization $\eta_\rho$ is $1/0.3125 = 3.2$.

**Example 2.** Now consider an ensemble $\rho$ of which 51% of classifiers always output Label 1, and the other 49% classifiers always output Label 2.

- **Case 1.** The polarization $\eta_\rho$ is now 0, the same as in Example 1.

- **Case 2.** The polarization $\eta_\rho$ is $0.5/0.2501 \approx 2$, which is larger than 1.6 in Example 1.

- **Case 3.** The polarization $\eta_\rho$ is now $1/0.2501 \approx 4$, which is larger than 3.2 in Example 1.

In addition, the following proposition draws a connection between polarization and the competence condition mentioned in Section 2.2. It states that the polarization of competent ensembles cannot be very large. The proof is deferred to Appendix A.2.

**Proposition 1.** *Competent ensembles are 2-polarized.*

Now we delve more into this new quantity. We introduce Theorem 1, which establishes (by means of concentration inequalities) an upper bound on the polarization $\eta_\rho$. The proof of Theorem 1 is deferred to Appendix B.1.

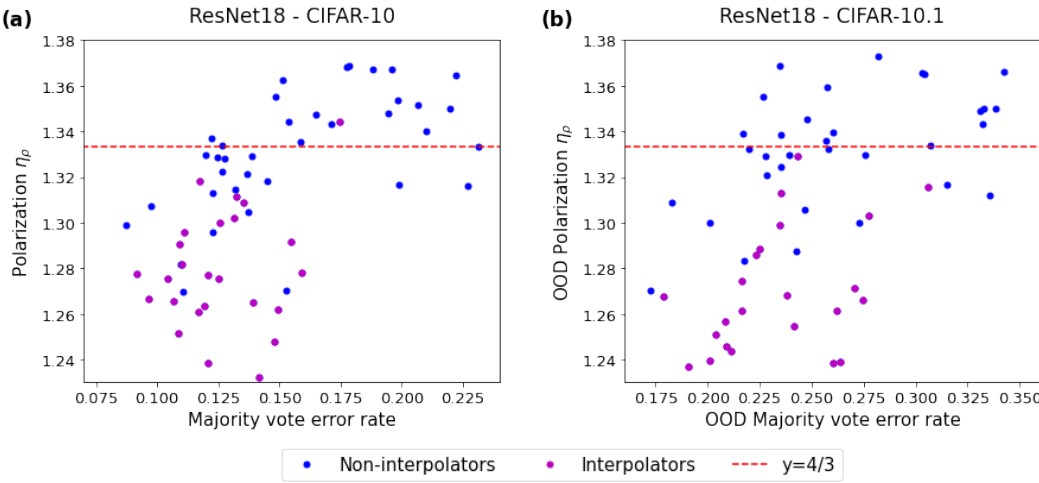

Figure 1: Polarizations $\eta_\rho$ obtained from ResNet18 trained on CIFAR-10 with various sets of hyper-parameters tested on **(a)** an out-of-sample CIFAR-10 and **(b)** an out-of-distribution dataset, CIFAR-10.1. Red dashed line indicates $y = 4/3$, a suggested value of polarization appears in Theorem 1 and Conjecture 1.

**Theorem 1.** *Let $\{(X_i, Y_i)\}_{i=1}^m$ be independent and identically distributed samples from $\mathcal{D}$ that are independent of an ensemble $\rho$. Then the polarization of the ensemble, $\eta_\rho$, satisfies*

$$\eta_\rho \leq \max\left\{\frac{4}{3}, \left(\frac{\sqrt{\frac{3}{8m}\log\frac{1}{\delta}} + \sqrt{\frac{3}{8m}\log\frac{1}{\delta} + 4SP}}{2S}\right)^2\right\}, \qquad (9)$$

*with probability at least $1 - \delta$, where $S = \frac{1}{m}\sum_{i=1}^m W_\rho^2(X_i, Y_i)$ and $P = \frac{1}{m}\sum_{i=1}^m \mathbb{1}(W_\rho(X_i, Y_i) > 1/2)$.f*

Surprisingly, in practice, $\eta_\rho = \frac{4}{3}$ appears to be a good choice for a wide variety of cases. See Figure 1 and Figure 2, which show the polarization $\eta_\rho$ obtained from VGG11 [SZ14], DenseNet40 [HLVDMW17], ResNet18, ResNet50 and ResNet101 [HZRS16] trained on CIFAR-10 [Kri09] with various hyperparameters choices. The trend does not deviate even when evaluated on an out-of-distribution dataset, CIFAR-10.1 [RRSS18, TFF08]. For more details on these empirical results, see Appendix C.

**Remark.** We emphasize that values for $\eta_\rho$ that are larger than $\frac{4}{3}$ does *not* contradict Theorem 1. This happens when the non-constant second term in (9) is larger than $\frac{4}{3}$, which is often the case for classifiers which are not interpolating (or, indeed, that underfit or perform poorly).

**Definition 2** (INTERPOLATING, [BHMM19]). *A classifier is **interpolating** if it achieves an accuracy of 100% on the training data.*

Putting Theorem 1 and the consistent empirical trend shown in Figure 2(b) together, we propose the following conjecture.

**Conjecture 1** (NEURAL POLARIZATION LAW). *The polarization of ensembles comprised of independently trained interpolating neural networks is smaller than $\frac{4}{3}$.*

## 4 Entropy-Restricted Ensembles

In this section, we first present an upper bound on the majority vote error rate, $L(h_\rho^{\mathrm{MV}})$, in Theorem 2, using our notion of polarization $\eta_\rho$ which we introduced and defined in the previous section. Then, we present Theorems 3 and 4 which are the main elements in obtaining tighter upper bounds on $L(h_\rho^{\mathrm{MV}})$. Figure 3 shows our proposed bound offers a significant improvement over state-of-the-art results. The new upper bounds are inspired from the fact that classifier prediction probabilities tend to concentrate on a small number of labels, rather than be uniformly spread over all the possible labels.

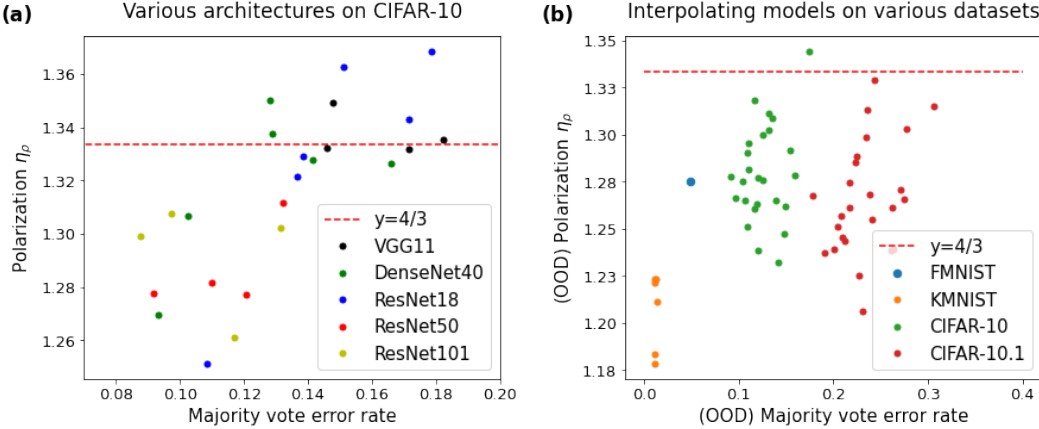

Figure 2: Polarization $\eta_\rho$ obtained **(a)** from various architectures trained on CIFAR-10 and **(b)** only from interpolating classifiers trained on various datasets. Red dashed line indicates $y = 4/3$. In subplot **(b)**, we observe that the polarization of all interpolating models expect one are smaller than $4/3$, which aligns with Conjecture 1.

This is analogous to the phenomenon of *neural collapse* [Kot22]. As an example, in the context of a computer vision model, when presented with a photo of a dog, one might expect that a large portion of reasonable models might classify the photo as an animal other than a dog, but not as a car or an airplane.

We start by stating an upper bound on the majority vote error, $L(h_\rho^{\mathrm{MV}})$ as a function of polarization $\eta_\rho$. This upper bound is tighter (smaller) than the previous bound in inequality (6) when the polarization is lower than 2, which is the case for competent ensembles. The proof is deferred to Appendix B.2.

**Theorem 2.** *For an ensemble $\rho$ of $K$-class classifiers,*

$$L(h_\rho^{\mathrm{MV}}) \leq \frac{2\eta_\rho(K-1)}{K}\left(\mathbb{E}_\rho[L(h)] - \frac{1}{2}\mathbb{E}_{\rho^2}[D(h,h')]\right),$$

*where $\eta_\rho$ is the polarization of the ensemble $\rho$.*

Based on the upper bound stated in Theorem 2, we add a restriction on the entropy of constituent classifiers to obtain Theorem 3. The theorem provides a tighter scalable bound that does *not* have explicit dependency on the total number of labels, with a small cost in terms of the entropy of constituent classifiers. The proof of Theorem 3 is deferred to Appendix B.3.

**Theorem 3.** *Let $\rho$ be any $\eta$-polarized ensemble of $K$-class classifiers that satisfies $\mathbb{P}_\rho(h(x) \notin A(x)) \leq \Delta$, where $y \in A(x) \subset [K]$ and $|A(x)| \leq M$, for all data points $(x,y) \in \mathcal{D}$. Then, we have*

$$L(h_\rho^{\mathrm{MV}}) \leq \frac{2\eta(M-1)}{M}\left[\left(1 + \frac{\Delta}{M-1}\right)\mathbb{E}_\rho[L(h)] - \frac{1}{2}\mathbb{E}_{\rho^2}[D(h,h')]\right].$$

While Theorem 3 might provide a tighter bound than prior work, coming up with pairs $(M, \Delta)$ that satisfy the constraint is not an easy task. This is not an issue for a discrete ensemble, however. If $\rho$ is a discrete distribution of $N$ classifiers, then we observe that the assumption of Theorem 3 must always hold with $(M, \Delta) = (N+1, 0)$. We state this as the following corollary.

**Corollary 1** (FINITE ENSEMBLE). *For an ensemble $\rho$ that is a weighted sum of $N$ classifiers, we have*

$$L(h_\rho^{\mathrm{MV}}) \leq \frac{2\eta_\rho N}{N+1}\left(\mathbb{E}_\rho[L(h)] - \frac{1}{2}\mathbb{E}_{\rho^2}[D(h,h')]\right), \tag{10}$$

*where $\eta_\rho$ is the polarization of the ensemble $\rho$.*

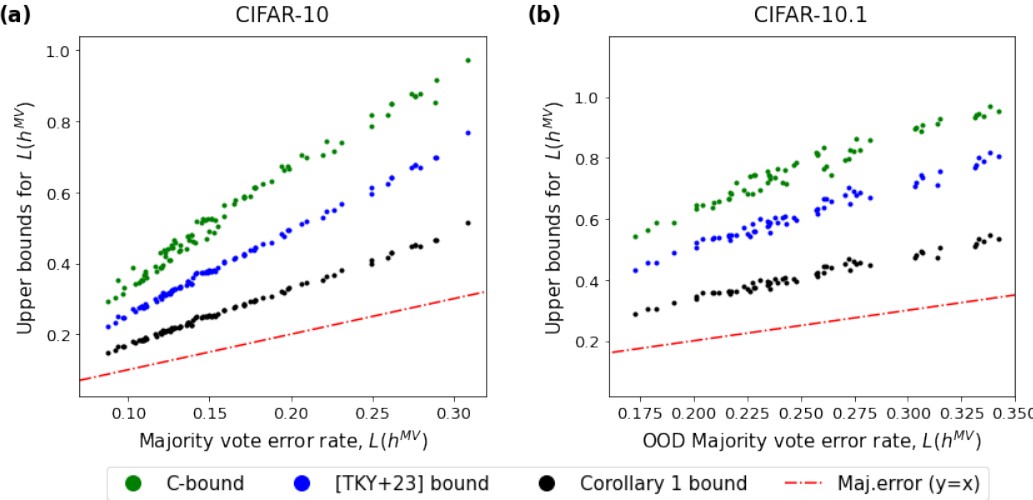

Figure 3: Comparing our new bound from Corollary 1 (colored black), which is the right hand side of inequality (10), with bounds from previous studies. Green corresponds to the C-bound in inequality (3), and blue corresponds to the right hand side of inequality (6). ResNet18, ResNet50, ResNet101 models with various sets of hyperparameters are trained on CIFAR-10 then tested on **(a)** the out-of-sample CIFAR-10, **(b)** an out-of-distribution dataset, CIFAR-10.1.

See Figure 3, which provides empirical results that compare the bound in Corollary 1 with the C-bound in inequality (3), and with inequality (6) proposed in [TKY$^+$24]. We can observe that the new bound in Corollary 1 is strictly tighter than the others. For more details on these empirical results, see Appendix C.

Although the bound in Corollary 1 is tighter than the bounds from previous studies, it's still not tight enough to use it as an estimator for $L(h_\rho^{\mathrm{MV}})$. In the following theorem, we use a stronger condition on the entropy of an ensemble to obtain a tighter bound. The proof is deferred to Appendix B.4.

**Theorem 4.** *For any $\eta$-polarized ensemble $\rho$ that satisfies*

$$\frac{1}{2}\mathbb{E}_{\mathcal{D}}\left[\mathbb{P}_{\rho^2}\left(h(X) \neq Y, h'(X) \neq Y, h(X) \neq h'(X)\right)\right] \leq \varepsilon \, \mathbb{E}_{\mathcal{D}}\left[\mathbb{P}_\rho\left(h(X) \neq Y\right)\right], \qquad (11)$$

*we have*

$$L(h_\rho^{\mathrm{MV}}) \leq \eta \left[(1+\varepsilon)\,\mathbb{E}_\rho[L(h)] - \frac{1}{2}\mathbb{E}_{\rho^2}[D(h,h')]\right].$$

The condition (11) can be rephrased as follows: compared to the error $\mathbb{P}_\rho(h(x) \neq y)$, the entropy of the distribution of wrong predictions is small, and it is concentrated on a small number of labels. A potential problem is that one must know or estimate the smallest possible value of $\varepsilon$ in advance. At least, we can prove that $\varepsilon = \frac{K-2}{2(K-1)}$ always satisfies the condition (11) for an ensemble of $K$-class classifiers. The proof is deferred to Appendix B.4.

**Corollary 2.** *For any $\eta$-polarized ensemble $\rho$ of K-class classifiers, we have*

$$L(h_\rho^{\mathrm{MV}}) \leq \eta \left[\left(1 + \frac{K-2}{2(K-1)}\right)\mathbb{E}_\rho[L(h)] - \frac{1}{2}\mathbb{E}_{\rho^2}[D(h,h')]\right].$$

Naturally, this $\varepsilon$ is not good enough for our goal. We discuss more on how to estimate the smallest possible value of $\varepsilon$ in the following section.

## 5  A Universal Law for Ensembling

In this section, our goal is to predict the majority vote error rate of an ensemble with large number of classifiers by just using information we can obtain from an ensemble with a small number, e.g., three,

of classifiers. Among the elements in the bound in Theorem 4,

$$\eta \left[ (1 + \varepsilon) \, \mathbb{E}_\rho[L(h)] - \frac{1}{2} \mathbb{E}_{\rho^2}[D(h, h')] \right],$$

we plug in $\eta = \frac{4}{3}$ as a result of Theorem 1; and since $\mathbb{E}_\rho[L(h)]$ is invariant to the number of classifiers, it remains to predict the behavior of $\mathbb{E}_{\rho^2}[D(h, h')]$ and the smallest possible value of $\varepsilon$, $\varepsilon_\rho = \frac{\mathbb{E}_\mathcal{D} \left[ \mathbb{P}_{\rho^2} \left( h(X) \neq Y, h'(X) \neq Y, h(X) \neq h'(X) \right) \right]}{2 \mathbb{E}_\mathcal{D} \left[ \mathbb{P}_\rho (h(X) \neq Y) \right]}$. Since the denominator $\mathbb{E}_\mathcal{D} \left[ \mathbb{P}_\rho (h(X) \neq Y) \right] = \mathbb{E}_\rho[L(h)]$ is invariant to the number of classifiers, and the numerator resembles the disagreement between classifiers, $\varepsilon_\rho$ is expected to follow a similar pattern as $\mathbb{E}_{\rho^2}[D(h, h')]$. Note that the numerator of $\varepsilon_\rho$ has the same form as the disagreement, differing by only one less label. Both are $V$-statistics that can be expressed as a multiple of a $U$-statistic, as shown in equation (12). In the next theorem, we show that the disagreement for a finite number of classifiers can be expressed as the sum of a hyperbolic curve and an unbiased random walk. Here, $[x]$ denotes the greatest integer less than or equal to $x$ and $\mathcal{D}[0, 1]$ is the Skorokhod space on $[0, 1]$ (see Appendix B.5).

**Theorem 5.** *Let $\rho_N$ denote an empirical distribution of $N$ independent classifiers $\{h_i\}_{i=1}^N$ sampled from a distribution $\rho$ and $\sigma_1^2 = \mathsf{Var}_{h \sim \rho}(\mathbb{E}_{h' \sim \rho} \mathbb{P}_\mathcal{D}(h(X) \neq h'(X)))$. Then, there exists $D_\infty > 0$ such that*

$$\mathbb{E}_{(h, h') \sim \rho_N^2}[D(h, h')] = \left( 1 - \frac{1}{N} \right) \left( D_\infty + \frac{2}{\sqrt{N}} Z_N \right),$$

*where $\mathbb{E}Z_N = 0$, $\mathsf{Var}Z_N \to \sigma_1^2$ and $\{\frac{\sqrt{t}}{\sigma_1} Z_{[Nt]}\}_{t \in [0,1]}$ converges weakly to a standard Wiener process in $\mathcal{D}[0, 1]$ as $N \to \infty$.*

*Proof.* Let $\Phi(h_i, h_j) = \mathbb{P}_\mathcal{D}(h_i(X) \neq h_j(X))$. We observe that

$$\frac{N^2}{N(N-1)} \mathbb{E}_{(h, h') \sim \rho_N^2}[D(h, h')] = \frac{1}{N(N-1)} \sum_{i,j=1}^N \mathbb{P}_\mathcal{D}(h_i(X) \neq h_j(X))$$

$$= \frac{1}{N(N-1)} \sum_{i,j=1}^N \Phi(h_i, h_j) \underset{\substack{\Phi: \text{ symmetric} \\ \Phi(h_i, h_i) = 0}}{=} \frac{2}{N(N-1)} \sum_{1 \leq i < j \leq N} \Phi(h_i, h_j) =: U_N, \quad (12)$$

which is a $U$-statistic with the kernel function $\Phi$. Let $\Phi_0 = \mathbb{E}_{(h, h') \sim \rho^2} \Phi(h, h')$.

The invariance principle of $U$-statistics (Theorem 7 in Appendix B.5) states that the process $\xi_N = (\xi_N(t), t \in [0, 1])$, defined by $\xi_N(\frac{k}{N}) = \frac{k}{2\sqrt{N \sigma_1^2}}(U_k - \Phi_0)$ and $\xi_N(t) = \xi_N(\frac{[Nt]}{N})$, converges weakly to a standard Wiener process in $\mathcal{D}[0, 1]$ as $N \to \infty$, since $\sigma_1^2 = \mathsf{Var}_{h \sim \rho} \mathbb{E}_{h' \sim \rho} \Phi(h, h')$. Therefore, $U_N$ converges in probability as $N \to \infty$ to $D_\infty := \Phi_0$.

Letting $Z_N = \sigma_1 \xi_N(1) = \frac{\sqrt{N}}{2}(U_N - D_\infty)$, we can express $U_N$ as $U_N = D_\infty + \frac{2}{\sqrt{N}} Z_N$, with $\mathbb{E}Z_N = 0$ and $\mathsf{Var}Z_N \to \sigma_1^2$. Since $\frac{\sqrt{t}}{\sigma_1} Z_{[Nt]} = \sqrt{\frac{Nt}{[Nt]}} \xi_N(\frac{[Nt]}{N}) = \sqrt{\frac{Nt}{[Nt]}} \xi_N(t)$, it follows by Slutsky's Theorem that $\{\frac{\sqrt{t}}{\sigma_1} Z_{[Nt]}\}_{t \in [0,1]}$ converges weakly to a standard Wiener process in $\mathcal{D}[0, 1]$ as $N \to \infty$. $\quad\square$

Theorem 5 suggests that the disagreement within $N$ classifiers, $\mathbb{E}_{\rho_N^2}[D(h, h')]$, can be approximated as $\frac{N-1}{N} D_\infty$. From the disagreement within $M(\ll N)$ classifiers, $D_\infty$ can be approximated as $\frac{M}{M-1} \mathbb{E}_{\rho_M^2}[D(h, h')]$, and therefore we get

$$\mathbb{E}_{\rho_N^2}[D(h, h')] \approx \frac{N-1}{N} \cdot \frac{M}{M-1} \mathbb{E}_{\rho_M^2}[D(h, h')]. \quad (13)$$

Assume that we have three classifiers sampled from $\rho$. We denote the average error rate, the disagreement, and the $\varepsilon_\rho$ from these three classifiers by $\mathbb{E}_3[L(h)]$, $\mathbb{E}_3[D(h, h')]$, and $\varepsilon_3$, respectively. Then, from Theorem 4 and approximation (13) (which applies to both disagreement and $\varepsilon_\rho$), we

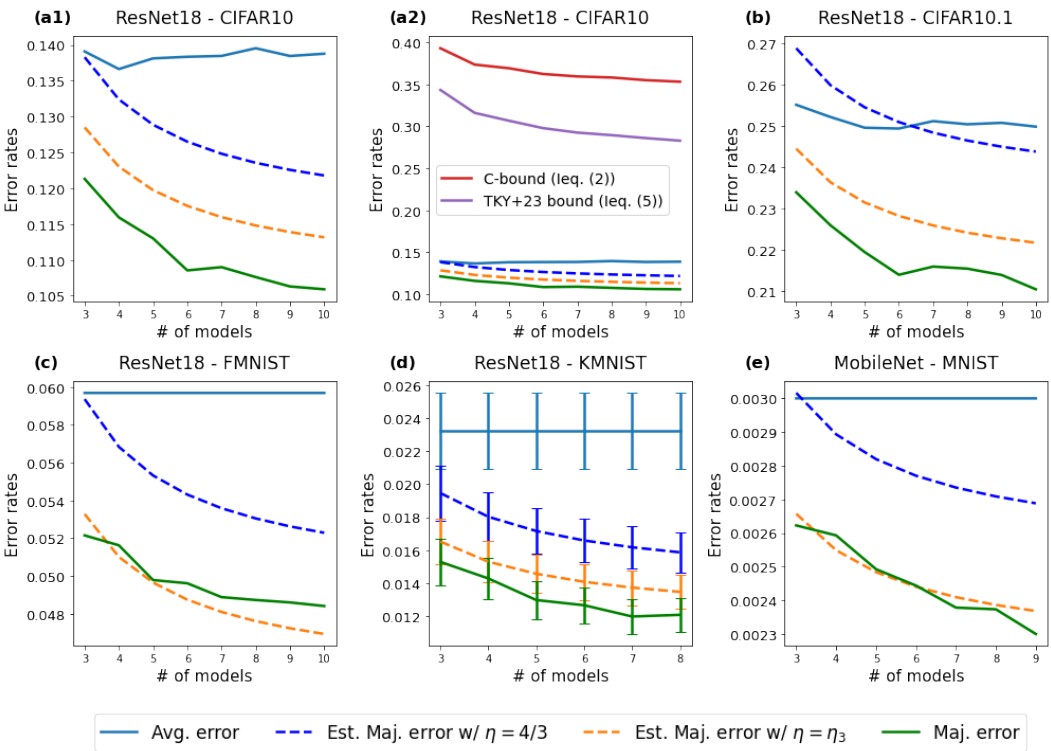

Figure 4: Comparing the estimated (extrapolated) majority vote error rates in equation (14) (blue-dashed lines) and (15) (orange-dashed lines) with the true majority vote error (green solid line) for each number of classifiers. The solid sky-blue line corresponds to the average error rate of constituent classifiers. Subplots **(a1)**, **(b)**, **(c)**, **(d)**, **(e)** show the results from different pairs of (classification model, dataset). Subplot **(a2)** overlays the right hand side of inequality (3) (C-bound, colored red) and inequality (6) ([TKY+24] bound, colored purple) on the subplot **(a1)**. These two quantities from previous studies are much larger compared to the average error rate. We see the same pattern for other (architecture, dataset) pairs, which we therefore omit from the plot. For more details on these empirical results, see Appendix C.

estimate the majority vote error rate of $N$ classifiers from $\rho$ as the following:

$$L(h_\rho^{\mathrm{MV}}) \lessapprox \frac{4}{3} \left[ \left( 1 + \frac{N-1}{N} \cdot \frac{3}{2} \cdot \varepsilon_3 \right) \mathbb{E}_3[L(h)] - \frac{N-1}{N} \cdot \frac{3}{2} \cdot \frac{1}{2} \mathbb{E}_3[D(h,h')] \right]$$

$$= \frac{4}{3} \left[ \mathbb{E}_3[L(h)] + \frac{3(N-1)}{2N} \left( \varepsilon_3 \mathbb{E}_3[L(h)] - \frac{1}{2} \mathbb{E}_3[D(h,h')] \right) \right]. \tag{14}$$

Alternatively, we can use the polarization measured from three classifiers, $\eta_3$, instead of $\eta = \frac{4}{3}$, to obtain:

$$L(h_\rho^{\mathrm{MV}}) = \eta_3 \left[ \mathbb{E}_3[L(h)] + \frac{3(N-1)}{2N} \left( \varepsilon_3 \mathbb{E}_3[L(h)] - \frac{1}{2} \mathbb{E}_3[D(h,h')] \right) \right]. \tag{15}$$

Figure 4 presents empirical results that compare the estimated (extrapolated) majority vote error rates in equations (14) and (15) with the true majority vote error for each number of classifiers. ResNet18 models are tested on four different dataset: CIFAR-10, CIFAR-10.1, Fashion-MNIST [XRV17] and Kuzushiji-MNIST [CBIK+18] where the models are trained on the corresponding train data. MobileNet [How17] is trained and tested on the MNIST [Den12] dataset. Not only do the estimators show significant improvement compared to the bounds introduced in Section 2.2, we observe that the estimators are very close to the actual majority vote error rate; and thus the estimators have practical usages, unlike the bounds from previous studies. In Figure 4(a2), existing bounds (3) and (6) are much larger compared to the average error rate. This is also the case for (architecture, dataset) pairs of other subplots.

## 6 Discussion and Conclusion

This work addresses the question: how does the majority vote error rate change according to the number of classifiers? While this is an age-old question, it is one that has received renewed interest in recent years. On the journey to answering the question, we introduce several new ideas of independent interest. (1) We introduced the polarization $\eta_\rho$, of an ensemble of classifiers. This notion plays an important role throughout this paper and appears in every upper bound presented. Although Theorem 1 gives some insight into polarization, our conjectured neural polarization law (Conjecture 1) is yet to be proved or disproved, and it provides an exciting avenue for future work. (2) We proposed two classes of ensembles whose entropy is restricted in different ways. Without these constraints, there will always be examples that saturate even the least useful majority vote error bounds. We believe that accurately describing how models behave in terms of the entropy of their output is key to precisely characterizing the behavior of majority vote, and likely other ensembling methods.

Throughout this paper, we have theoretically and empirically demonstrated that polarization is fairly invariant to the hyperparameters and architecture of classifiers. We also proved a tight bound for majority vote error, under an assumption with another quantity $\varepsilon$, and we presented how the components of this tight bound behave according to the number of classifiers. Altogether, we have sharpened bounds on the majority vote error to the extent that we are able to identify the trend of majority vote error rate in terms of number of classifiers.

We close with one final remark regarding the metrics used to evaluate an ensemble. Majority vote error rate is the most common and popular metric used to measure the performance of an ensemble. However, it seems unlikely that a practitioner would consider an ensemble to have performed adequately if the majority vote conclusion was correct, but was only reached by a relatively small fraction of the classifiers. With the advent of large language models, it is worth considering whether the majority vote error rate is still as valuable. The natural alternative in this regard is the probability $\mathbb{P}_\rho(W_\rho > 1/2)$, that is, the probability that at least half of the classifiers agree on the correct answer. This quantity is especially well-behaved, and it frequently appears in our proofs. (Indeed, every bound presented in this work serves as an upper bound for $\mathbb{P}_\rho(W_\rho > 1/2)$.) We conjecture that this quantity is useful much more generally.

**Acknowledgements.** We would like to thank the DOE, IARPA, NSF, and ONR for providing partial support of this work.

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

# A   More discussion on competence

In this section, we delve more into the competence condition that was introduced in [TKY+24]. We explore in which cases the competence condition might not work and how to overcome these issues. We discuss a few milder versions of competence that are enough for bounds (5) and (6) to hold. Then we discuss how to check whether these weaker competence conditions hold in practice, with or without a separate validation set. We start by formally stating the original competence condition.

**Definition 3** (Competence, [TKY+24])**.** *The ensemble $\rho$ is* **competent** *if for every $0 \leq t \leq 1/2$,*

$$\mathbb{P}_{\mathcal{D}}(W_\rho \in [t, 1/2)) \geq \mathbb{P}_{\mathcal{D}}(W_\rho \in [1/2, 1-t]). \tag{16}$$

## A.1   Cases when competence fails

One tricky part in the definition of competence is that it requires inequality (16) to hold for **every** $0 \leq t \leq 1/2$. In case $t = 1/2$, the inequality becomes

$$0 \geq \mathbb{P}_{\mathcal{D}}(W_\rho = 1/2).$$

This is not a significant issue in the case that $\rho$ is a continuous distribution over classifiers, e.g., a Bayes posterior or a distribution over a parametric family $h_\theta$, as $\{W_\rho = 1/2\}$ would be a measure-zero set. In the case that $\rho$ is a discrete distribution over finite number of classifiers, however, $\mathbb{P}_{\mathcal{D}}(W_\rho = 1/2)$ is likely to be a positive quantity, in which case it can violate the competence condition.

That being said, $\{(x,y) \mid W_\rho(x,y) = 1/2\}$ represent tricky data points that deserves separate attention. This event can be divided into two cases: 1) all the classifiers that incorrectly made a prediction output the same label; or 2) incorrect predictions consist of multiple labels so that the majority vote outputs the true label. Among these two possibilities, the first case is troublesome. We denote such data points by $\text{TIE}(\rho, \mathcal{D})$:

$\text{TIE}(\rho, \mathcal{D}) :=$
$\quad \{(x,y) \mid \mathbb{P}_\rho(\mathbb{1}(h(x) = j)) = \mathbb{P}_\rho(\mathbb{1}(h(x) = y)) = 1/2 \text{ for true label } y \text{ and an incorrect label } j\}.$

In this case, the true label and an incorrect label are chosen by exactly the same $\rho-$weights of classifiers. An easy way to resolve this issue is to slightly tweak the weights. For instance, if $\rho$ is an equally weighted sum of two classifiers, we can change each of their weights to be $(1/2 + \epsilon, 1/2 - \epsilon)$, instead of $(1/2, 1/2)$. This change may seem manipulative, but it corresponds to a deterministic tie-breaking rule which prioritizes one classifier over the other, which is a commonly used tie-breaking rule.

**Definition 4** (Tie-free ensemble)**.** *An ensemble is* **tie-free** *if $\mathbb{P}_{\mathcal{D}}(TIE(\rho, \mathcal{D})) = 0$.*

**Proposition 2.** *An ensemble with a deterministic tie-breaking rule is tie-free.*

With such tweak to make the set $\text{TIE}(\rho, \mathcal{D})$ to be an empty set or a measure-zero set, we present a slightly milder condition that is enough for the bounds (5) and (6) to still hold.

**Definition 5** (Semi-competence)**.** *The ensemble $\rho$ is* **semi-competent** *if for every $0 \leq t < 1/2$,*

$$P(W_\rho \in [t, 1/2]) \geq \mathbb{P}(W_\rho \in (1/2, 1-t]). \tag{17}$$

Note that inequality (17) is a strictly weaker condition than inequality (16), and hence competence implies semi-competence. The converse is not true. An ensemble is semi-competent even if the point-wise error $W_\rho(X, Y) = 1/2$ on every data points, but such an ensemble is not competent.

**Theorem 6.** *For a tie-free ensemble and semi-competent ensemble $\rho$, $L(h_\rho^{\mathrm{MV}}) \leq \mathbb{E}_\rho[L(h)]$ and*

$$L(h_\rho^{\mathrm{MV}}) \leq \frac{4(K-1)}{K} \left( \mathbb{E}_\rho[L(h)] - \frac{1}{2} \mathbb{E}_{\rho^2}[D(h, h')] \right)$$

*holds in $K$-class classification setting.*

We provide the proof as a separate subsection below.

## A.2 Proof of Theorem 6 and Proposition 1

We start with the following lemma, which is a semi-competence version of Lemma 2 from [TKY$^+$24].

**Lemma 1.** *For a semi-competent ensemble $\rho$ and any increasing function $g$ satisfying $g(0) = 0$,*

$$\mathbb{E}_{\mathcal{D}}[g(W_\rho)\mathbb{1}_{W_\rho \leq 1/2}] \geq \mathbb{E}_{\mathcal{D}}[g(\widetilde{W}_\rho)\mathbb{1}_{\widetilde{W}_\rho < 1/2}],$$

*where $\widetilde{W}_\rho = 1 - W_\rho$.*

*Proof.* For every $x \in [0, 1]$, it holds that

$$\mathbb{P}_{\mathcal{D}}(W_\rho\mathbb{1}_{W_\rho \leq 1/2} \geq x) = \mathbb{P}_{\mathcal{D}}(W_\rho \in [x, 1/2])\,\mathbb{1}_{x \leq 1/2},$$
$$\mathbb{P}_{\mathcal{D}}(\widetilde{W}_\rho\mathbb{1}_{\widetilde{W}_\rho < 1/2} \geq x) = \mathbb{P}_{\mathcal{D}}(\widetilde{W}_\rho \in [x, 1/2))\,\mathbb{1}_{x \leq 1/2} = \mathbb{P}_{\mathcal{D}}(W_\rho \in (1/2, 1-x])\,\mathbb{1}_{x \leq 1/2}.$$

From the definition of semi-competence, this implies that $\mathbb{P}_{\mathcal{D}}(W_\rho\mathbb{1}_{W_\rho \leq 1/2} \geq x) \geq \mathbb{P}_{\mathcal{D}}(\widetilde{W}_\rho\mathbb{1}_{\widetilde{W}_\rho < 1/2} \geq x)$ for every $x \in [0, 1]$. Using the fact that $g(x\,\mathbb{1}_{x \leq c}) = g(x)\mathbb{1}_{x \leq c}$ for any increasing function $g$ with $g(0) = 0$, we obtain

$$\mathbb{P}_{\mathcal{D}}(h(W_\rho)\mathbb{1}_{W_\rho \leq 1/2} \geq x) \geq \mathbb{P}_{\mathcal{D}}(h(\widetilde{W}_\rho)\mathbb{1}_{\widetilde{W}_\rho < 1/2} \geq x).$$

Putting these together with a well-known equality $\mathbb{E}X = \int_0^\infty \mathbb{P}(X \geq x)\mathrm{d}x$ for a non-negative random variable $X$ proves the lemma. □

Now we use Lemma 1 and Theorem 2 to prove Theorem 6.

*Proof of Theorem 6.* Applying Lemma 1 with $g(x) = 2x^2$ gives,

$$\mathbb{E}_{\mathcal{D}}[2W_\rho^2\mathbb{1}_{W_\rho \leq 1/2}] \geq \mathbb{E}_{\mathcal{D}}[2\widetilde{W}_\rho^2\mathbb{1}_{\widetilde{W}_\rho < 1/2}] = \mathbb{E}_{\mathcal{D}}[(2 - 4W_\rho + 2W_\rho^2)\mathbb{1}_{W_\rho > 1/2}]. \tag{18}$$

Putting this together with the following decomposition of $\mathbb{E}_{\mathcal{D}}[2W_\rho^2]$ shows that the ensemble $\rho$ is 2-polarized:

$$\begin{aligned}
\mathbb{E}_{\mathcal{D}}[2W_\rho^2] &\geq \mathbb{E}_{\mathcal{D}}[2W_\rho^2\mathbb{1}_{W_\rho > 1/2}] + \mathbb{E}_{\mathcal{D}}[2W_\rho^2\mathbb{1}_{W_\rho \leq 1/2}] \\
&\underset{(18)}{\geq} \mathbb{E}_{\mathcal{D}}[2W_\rho^2\mathbb{1}_{W_\rho > 1/2}] + \mathbb{E}_{\mathcal{D}}[(2 - 4W_\rho + 2W_\rho^2)\mathbb{1}_{W_\rho > 1/2}] \\
&\geq \mathbb{E}_{\mathcal{D}}[(1 - 2W_\rho)^2\mathbb{1}_{W_\rho > 1/2}] + \mathbb{P}_{\mathcal{D}}(W_\rho > 1/2) \\
&\geq \mathbb{P}_{\mathcal{D}}(W_\rho > 1/2).
\end{aligned} \tag{19}$$

Therefore, applying Theorem 2 with constant $\eta = 2$ concludes the proof. □

We also state the following proof of Proposition 1 for completeness.

*Proof of Proposition 1.* Inequality (19) with Lemma 3 proves the proposition. □

## A.3 Example that the bound (6) is tight

Here, we provide a combination of $(\rho, \mathcal{D})$ of which $L(h_\rho^{\mathrm{MV}})$ is arbitrarily close to the bound.

Consider, for each feature $x$, that exactly $(1 - \epsilon)$ fraction of classifiers predict the correct label, and that the remaining $\epsilon$ fraction of classifiers predict a wrong label. In this case, $L(h_\rho^{\mathrm{MV}}) = 0$, $\mathbb{E}_\rho[L(h)] = \epsilon$, and $\mathbb{E}_{\rho^2}[D(h, h')] = 2\epsilon(1 - \epsilon)$. Hence, the upper bound (6) is $\frac{4(K-1)}{K}\epsilon^2$, which can be arbitrarily close to 0.

# B Proofs of our main results

In this section, we provide proofs for our main results.

## B.1 Proof of Theorem 1

We start with the following lemma which shows the concentration of a linear combination of $W_\rho^2$ and $\mathbb{1}(W_\rho > 1/2)$.

**Lemma 2.** *For sampled data points $\{(X_i, Y_i)\}_{i=1}^m \sim \mathcal{D}$, define $Z_2 := \sum_{i=1}^m W_\rho^2(X_i, Y_i)$ and $Z_0 := \sum_{i=1}^m \mathbb{1}(W_\rho(X_i, Y_i) > 1/2)$. The ensemble $\rho$ is $\eta$-polarized with probability at least $1 - \delta$ if*

$$\frac{1}{m}(\eta Z_2 - Z_0) > \sqrt{\frac{\max\{\frac{3\eta}{4}, 1\}}{2m} \log \frac{1}{\delta}}. \tag{20}$$

*Proof.* Let $Z_{2i} = W_\rho^2(X_i, Y_i)$ and $Z_{0i} = \mathbb{1}(W_\rho(X_i, Y_i) > 1/2)$. Observe that $\eta Z_{2i} - Z_{0i}$ always takes a value between $[\frac{\eta}{4} - 1, \max\{\frac{\eta}{4}, \eta - 1\}]$ since $W_\rho(X_i, Y_i) \in [0, 1]$. This implies that $\eta Z_{2i} - Z_{0i}$s are i.i.d. sub-Gaussian random variable with parameter $\sigma = \max\{\frac{3\eta}{4}, 1\}/2$.

By letting $A_2 = \mathbb{E}[\eta W_\rho^2 - \mathbb{1}(W_\rho \geq 1/2)]$ and using the Hoeffding's inequality, we obtain

$$\frac{1}{m}(\eta Z_2 - Z_0) - A_2 \leq \sqrt{\frac{\max\{\frac{3\eta}{4}, 1\}}{2m} \log \frac{1}{\delta}}$$

with probability at least $1 - \delta$.

Therefore, $\rho$ is $\eta$-polarized with probability at least $1 - \delta$ if

$$\frac{1}{m}(\eta Z_2 - Z_0) > \sqrt{\frac{\max\{\frac{3\eta}{4}, 1\}}{2m} \log \frac{1}{\delta}}.$$

$\square$

Now we use Lemma 2 to prove Theorem 1.

*Proof of Theorem 1.* Observe that $S = \frac{1}{m} Z_2$, $P = \frac{1}{m} Z_0$, and thus $\frac{1}{m}(\eta Z_2 - Z_0) = \eta S - P$. For $\eta \geq \frac{4}{3}$, the lower bound in Lemma 2 is simply $\sqrt{\frac{3\eta}{8m} \log \frac{1}{\delta}}$, and the inequality (20) can be viewed as a quadratic inequality in terms of $\sqrt{\eta}$. From quadratic formula, we know that

$$\text{if} \quad \sqrt{\eta} > \frac{\sqrt{\frac{3\eta}{8m} \log \frac{1}{\delta}} + \sqrt{\frac{3\eta}{8m} \log \frac{1}{\delta} + 4SP}}{2S}, \quad \text{then} \quad \eta S - P - \sqrt{\frac{3\eta}{8m} \log \frac{1}{\delta}} > 0.$$

Putting this together with Lemma 2 proves the theorem:

$$\eta \geq \max\left\{\frac{4}{3}, \left(\frac{\sqrt{\frac{3}{8m} \log \frac{1}{\delta}} + \sqrt{\frac{3}{8m} \log \frac{1}{\delta} + 4SP}}{2S}\right)^2\right\} \tag{21}$$

$$\Rightarrow \quad \eta S - P > \sqrt{\frac{3\eta}{8m} \log \frac{1}{\delta}} \quad \underset{\text{Lemma 2}}{\Rightarrow} \quad \rho \text{ is } \eta\text{-polarized w.p. } 1 - \delta,$$

and thus the polarization $\eta_\rho$, the smallest $\eta$ such that $\rho$ is $\eta$-polarized, is upper bounded by the right hand side of inequality (21). $\square$

## B.2 Proof of Theorem 2

We start by proving the following lemma which relates the error rate of the majority vote, $L(h_\rho^{\mathrm{MV}})$, with the point-wise error rate, $W_\rho$, using Markov's inequality. In general, $L(h_\rho^{\mathrm{MV}}) \leq \mathbb{P}_\mathcal{D}(W_\rho \geq 1/2)$ is true for any ensemble $\rho$. We prove a tighter version of this. The difference between the two can be non-negligible when dealing with an ensemble with finite number of classifiers. Refer to Appendix A.1 and Definition 4 for more details regarding this difference and tie-free ensembles.

**Lemma 3.** *For a tie-free ensemble $\rho$, we have the inequality $L(h_\rho^{\mathrm{MV}}) \leq \mathbb{P}_\mathcal{D}(W_\rho > 1/2)$.*

*Proof.* For given feature $x$, $W_\rho \leq 1/2$ implies that more than or exactly $\rho-$weighted half of the classifiers outputs the true label. Since the ensemble $\rho$ is tie-free, $h_\rho^{\mathrm{MV}}$ outputs the true label if $W_\rho \leq 1/2$. Therefore, $\{(x, y) \mid W_\rho(x, y) \leq 1/2\} \subset \{(x, y) \mid h_\rho^{\mathrm{MV}}(x) = y\}$. Applying $\mathbb{P}_\mathcal{D}$ on the both sides proves the lemma. $\square$

The following lemma appears as Lemma 2 in [MLIS20]. This lemma draws the connection between the point-wise error rate, $W_\rho$ and the tandem loss, $\mathbb{E}_{\rho^2}[L(h, h')]$.

**Lemma 4.** *The equality $\mathbb{E}_\mathcal{D}[W_\rho{}^2] = \mathbb{E}_{\rho^2}[L(h, h')]$ holds.*

The next lemma appears as Lemma 4 in [TKY+24]. This lemma provides an upper bound on the tandem loss, $\mathbb{E}_{\rho^2}[L(h, h')]$, in terms of the average error rate, $\mathbb{E}_\rho[L(h)]$, and the average disagreement, $\mathbb{E}_{\rho^2}[D(h, h')]$.

**Lemma 5.** *For the $K$-class problem,*

$$\mathbb{E}_{\rho^2}[L(h, h')] \leq \frac{2(K-1)}{K} \left( \mathbb{E}_\rho[L(h)] - \frac{1}{2} \mathbb{E}_{\rho^2}[D(h, h')] \right).$$

Now we use these results to prove Theorem 2.

*Proof of Theorem 2.* Putting Lemmas 3, 4, and 5 and the definition of the polarization together proves the theorem:

$$L(h_\rho^{\mathrm{MV}}) \underset{\text{Lemma 3}}{\leq} \mathbb{P}_\mathcal{D}(W_\rho > 1/2) \underset{\text{polarization}}{\leq} \eta_\rho \, \mathbb{E}_\mathcal{D}[W_\rho^2]$$

$$\underset{\text{Lemma 4}}{=} \eta_\rho \, \mathbb{E}_{\rho^2}[L(h, h')] \underset{\text{Lemma 5}}{=} \frac{2\eta_\rho(K-1)}{K} \left( \mathbb{E}_h[L(h)] - \frac{1}{2} \mathbb{E}_{h,h'}[D(h, h')] \right).$$

$\square$

## B.3    Proof of Theorem 3

We start with a lemma which is a corollary of *Newton's inequality*.

**Lemma 6.** *For any collection of probabilities $p_1, \ldots, p_n$, the following inequality holds.*

$$\sum_{1 \leq i < j \leq n} p_i p_j \leq \frac{n-1}{2n} \left( \sum_{i=1}^n p_i \right)^2.$$

*Proof.* Newton's inequality states that

$$\frac{e_2}{\binom{n}{2}} \leq \left( \frac{e_1}{n} \right)^2 \qquad \text{where} \quad e_1 = \sum_{i=1}^n p_i \quad \text{and} \quad e_2 = \sum_{1 \leq i < j \leq n} p_i \, p_j.$$

Rearranging the terms gives the lemma. $\square$

Now we use this and the previous lemmas to prove Theorem 3.

*Proof of Theorem 3.* From Lemma 3, Lemma 4, and the definition of $\eta$-polarized ensemble, we have the following relationship between $L(h_\rho^{\mathrm{MV}})$ and $\mathbb{E}_{\rho^2}[L(h, h')]$:

$$L(h_\rho^{\mathrm{MV}}) \underset{\text{Lem. 3}}{\leq} \mathbb{P}_\mathcal{D}(W_\rho > 1/2) \underset{\eta\text{-polarized}}{\leq} \eta \, \mathbb{E}_\mathcal{D}[W_\rho{}^2] \underset{\text{Lemma 4}}{=} \eta \, \mathbb{E}_{\rho^2}[L(h, h')]. \tag{22}$$

From this, it suffices to prove that $h_\alpha \, \mathbb{E}_{\rho^2}[L(h, h')]$ is smaller than the upper bound in the theorem. First, observe the following decomposition of $\mathbb{E}_{\rho^2}[L(h, h')]$:

$$\mathbb{E}_{\rho^2}[L(h, h')] = \mathbb{E}_\mathcal{D}\left[ \mathbb{P}_\rho(h(X) \neq Y)^2 \right] = \mathbb{E}_\mathcal{D}\left[ \mathbb{P}_\rho(h(X) \neq Y) - \mathbb{P}_{\rho^2}(h(X) \neq Y, h'(X) = Y) \right]. \tag{23}$$

For any predictor mapping into $K$ classes, let $y$ denote the true label for an input $x$. Now we derive a lower bound of $\mathbb{P}_{\rho^2}(h(X) \neq Y, h'(X) = Y)$ using the following decomposition of $\mathbb{P}_{\rho^2}(h(x) \neq h'(x))$:

$$\frac{1}{2}\mathbb{P}_{\rho^2}(h(x) \neq h'(x))$$

$$= \mathbb{P}_{\rho^2}(h(x) \neq y, h'(x) = y) + \sum_{\substack{i \notin A(x) \\ j \in A(x)\setminus\{y\}}} p_i p_j, + \sum_{\substack{i,j \in A(x)\setminus\{y\} \\ i<j}} p_i p_j + \sum_{\substack{i,j \notin A(x) \\ i<j}} p_i p_j,$$

where $p_i := p_i(x) = \mathbb{P}_{\rho}(h(x) = i)$. We let $\Delta_x := \mathbb{P}_{\rho}(h(x) \notin A(x))$ and apply Lemma 6 to the last two terms:

$$\frac{1}{2}\mathbb{P}_{\rho^2}(h(x) \neq h'(x))$$

$$= \mathbb{P}_{\rho^2}(h(x) \neq y, h'(x) = y) + \Delta_x(1 - p_Y - \Delta_x) + \sum_{\substack{i,j \in A(x)\setminus\{y\} \\ i<j}} p_i p_j + \sum_{\substack{i,j \notin A(x) \\ i<j}} p_i p_j$$

$$\underset{\text{Lemma 6}}{\leq} \mathbb{P}_{\rho^2}(h(x) \neq y, h'(x) = y) + \Delta_x(1 - p_y - \Delta_x) + \frac{M-2}{2(M-1)}(1 - p_y - \Delta_x)^2 + \frac{K-M-1}{2(K-M)}\Delta_x^2.$$

Rearranging the terms and plugging $1 - p_Y = \mathbb{P}_{\rho}(h(x) \neq y)$ gives

$$\mathbb{P}_{\rho^2}(h(x) \neq y, h'(x) = y)$$

$$\geq \frac{1}{2}\mathbb{P}_{\rho^2}(h(x) \neq h'(x)) - \frac{\Delta_x}{M-1}\mathbb{P}_{\rho}(h(x) \neq y) - \frac{M-2}{2(M-1)}\mathbb{P}_{\rho}(h(x) \neq y)^2$$

$$+ \frac{K-1}{2(K-M)(M-1)}\Delta_x^2$$

$$\geq \frac{1}{2}\mathbb{P}_{\rho^2}(h(x) \neq h'(x)) - \frac{\Delta_x}{M-1}\mathbb{P}_{\rho}(h(x) \neq y) - \frac{M-2}{2(M-1)}\mathbb{P}_{\rho}(h(x) \neq y)^2$$

$$\geq \frac{1}{2}\mathbb{P}_{\rho^2}(h(x) \neq h'(x)) - \frac{\Delta}{M-1}\mathbb{P}_{\rho}(h(x) \neq y) - \frac{M-2}{2(M-1)}\mathbb{P}_{\rho}(h(x) \neq y)^2,$$

where the last inequality comes from the condition $\Delta_x := \mathbb{P}_{\rho}(h(x) \notin A(x)) \leq \Delta$. Putting this together with the equality (23) gives

$$\mathbb{E}_{\mathcal{D}}\left[\mathbb{P}_{\rho}(h(X) \neq Y)^2\right] \leq \left(1 + \frac{\Delta}{M-1}\right)\mathbb{E}_{\mathcal{D}}\left[\mathbb{P}_{\rho}(h(X) \neq Y)\right] - \frac{1}{2}\mathbb{E}_{\mathcal{D}}\left[\mathbb{P}_{\rho^2}(h(X) \neq h'(X))\right]$$

$$+ \frac{M-2}{2(M-1)}\mathbb{E}_{\mathcal{D}}\left[\mathbb{P}_{\rho}(h(X) \neq Y)^2\right],$$

which implies

$$\mathbb{E}_{\rho^2}[L(h, h')] = \mathbb{E}_{\mathcal{D}}\left[\mathbb{P}_{\rho}(h(X) \neq Y)^2\right]$$

$$\leq \frac{2(M-1)}{M}\left[\left(1 + \frac{\Delta}{M-1}\right)\mathbb{E}_{\mathcal{D}}\left[\mathbb{P}_{\rho}(h(X) \neq Y)\right] - \frac{1}{2}\mathbb{E}_{\mathcal{D}}\left[\mathbb{P}_{\rho^2}(h(X) \neq h'(X))\right]\right]$$

$$= \frac{2(M-1)}{M}\left[\left(1 + \frac{\Delta}{M-1}\right)\mathbb{E}_{\rho}[L(h)] - \frac{1}{2}\mathbb{E}_{\rho^2}[D(h, h')]\right].$$

Combining this with inequality (22) concludes the proof. $\qquad\square$

## B.4 Proof of Theorem 4 and Corollary 2

First, we prove Theorem 4 by decomposing the point-wise disagreement between constituent classifiers.

*Proof of Theorem 4.* The following decomposition of $\mathbb{P}_{\rho^2}(h(x) \neq h'(x))$ holds:

$$\frac{1}{2}\mathbb{P}_{\rho^2}(h(x) \neq h'(x)) = \mathbb{P}_{\rho^2}(h(x) \neq y, h'(x) = y) + \frac{1}{2}\mathbb{P}_{\rho^2}(h(x) \neq y, h'(x) = y, h(x) \neq h'(x)).$$

Applying $\mathbb{E}_{\mathcal{D}}$ to both sides and using the given condition (11), we obtain,

$$\frac{1}{2}\mathbb{E}_{\mathcal{D}}[\mathbb{P}_{\rho^2}(h(X) \neq h'(X))] \leq \mathbb{E}_{\mathcal{D}}\left[\mathbb{P}_{\rho^2}(h(X) \neq Y, h'(X) = Y)\right] + \varepsilon\,\mathbb{E}_{\mathcal{D}}[\mathbb{P}_{\rho}(h(X) \neq Y)].$$

The left hand side equals $\frac{1}{2}\mathbb{E}_{\rho^2}[D(h,h')]$, and the second term on the right hand side is simply $\varepsilon\,\mathbb{E}_{\rho}[L(h)]$. Hence, the inequality above can be rephrased as follows:

$$\frac{1}{2}\mathbb{E}_{\rho^2}[D(h,h')] - \varepsilon\,\mathbb{E}_{\rho}[L(h)] \leq \mathbb{E}_{\mathcal{D}}\left[\mathbb{P}_{\rho^2}(h(X) \neq Y, h'(X) = Y)\right]. \qquad (24)$$

Putting this together with the inequality (22) and the equality (23), gives

$$\begin{aligned}
L(h_\rho^{\mathrm{MV}}) &\underset{\text{Ineq.22}}{\leq} \eta\,\mathbb{E}_{\rho^2}[L(h,h')] \underset{\text{Eq.23}}{=} \eta\,\mathbb{E}_{\mathcal{D}}\left[\mathbb{P}_{\rho}(h(X) \neq Y) - \mathbb{P}_{\rho^2}(h(X) \neq Y, h'(X) = Y)\right] \\
&= \eta\left[\mathbb{E}_{\rho}[L(h)] - \mathbb{E}_{\mathcal{D}}\left[\mathbb{P}_{\rho^2}(h(X) \neq Y, h'(X) = Y)\right]\right] \\
&\underset{\text{Ineq.24}}{\leq} \eta\left[(1+\varepsilon)\mathbb{E}_{\rho}[L(h)] - \frac{1}{2}\mathbb{E}_{\rho^2}[D(h,h')]\right].
\end{aligned}$$

$\square$

Next, we use Lemma 6 to prove Corollary 2.

*Proof of Corollary 2.* Let $p_i := \mathbb{P}_{\rho}(h(x) = i)$ for $i \in [K]$, and let $y = K$ be the true label, without loss of generality. Then, we observe

$$\mathbb{P}_{\rho^2}(h(X) \neq Y, h'(X) \neq Y, h(X) \neq h'(X)) = \sum_{\substack{i,j=1 \\ i \neq j}}^{K-1} p_i p_j \quad \text{and} \quad \mathbb{P}_{\rho}(h(X) \neq Y) = \sum_{i=1}^{K-1} p_i.$$

Lemma 6 gives us the following:

$$\frac{\sum_{1 \leq i \neq j \leq K-1} p_i p_j}{2\sum_{1 \leq i \leq K-1} p_i} \leq \frac{\sum_{1 \leq i \neq j \leq K-1} p_i p_j}{2(\sum_{1 \leq i \leq K-1} p_i)^2} \leq \frac{\sum_{1 \leq i < j \leq K-1} p_i p_j}{(\sum_{1 \leq i \leq K-1} p_i)^2} \underset{\text{Lemma 6}}{\leq} \frac{K-2}{2(K-1)},$$

where the first inequality used the fact that $\sum_{i=1}^{K-1} p_i \leq 1$. Thus, $\varepsilon = \frac{K-2}{2(K-1)}$ satisfies the condition (11), and the result follows from Theorem 4. $\square$

### B.5 Invariance principle of $U$-statistics

In this subsection, we state the invariance principle of $U$-statistics, which plays a main role in the proof of Theorem 5. We note that this is a special case of an approximation of random walks (Theorem 23.14 in [Kal21]) combined with functional central limit theorem (Donsker's theorem). Here, $\mathcal{D}[0,1]$ is the Skorokhod space on $[0,1]$, which is the space of all real-valued right-continuous functions on $[0,1]$ equipped with the Skorokhod metric/topology (see Section 14 in [Bil13]).

**Theorem 7** (Theorem 5.2.1 in [KB13]). *Define a $U$-statistic $U_k = \binom{k}{2}^{-1}\sum_{1 \leq i < j \leq k}\Phi(h_i, h_j)$, the expectation of the kernel $\Phi$ as $\Phi_0 = \mathbb{E}_{(h,h') \sim \rho^2}\Phi(h,h')$ and the first-coordinate variance $\sigma_1^2 = \mathsf{Var}_{h \sim \rho}(g_1(h))$, where $g_1(h) = \mathbb{E}_{h' \sim \rho}\Phi(h', h)$. Let $\xi_n = (\xi_n(t), t \in [0,1])$, where*

$$\xi_n\left(\frac{k}{n}\right) = \frac{k(U_k - \Phi_0)}{2\sqrt{n\sigma_1^2}} \qquad \text{for} \quad k = 0, 1, ..., n-1,$$

*and $\xi_n(t) = \xi_n([nt]/n)$, with $[x]$ denoting the greatest integer less than or equal to $x$. Then, $\xi_n$ converges weakly in $\mathcal{D}[0,1]$ to a standard Wiener process as $n \to \infty$.*

## C  Details on our empirical results

In this section, we provide additional details on our empirical results.

## C.1 Trained classifiers

On CIFAR-10 [Kri09] train set with size $50,000$, the following models were trained with 100 epochs, learning rate starting with 0.05. For models trained with learning rate decay, we used learning rate 0.005 after epoch 50, and used 0.0005 after epoch 75. For following models, 5 classifiers are trained for each hyperparameter combination. Five classifiers differ in weight initialization and vary due to the randomized batches used during training.

- ResNet18, every combination (width, batch size) of
    - Width: $4, 8, 16, 32, 64, 128$
    - Batch size: $16, 128, 256, 1024$, with learning rate decay
      Additional batch size of $64, 180, 364$ for without learning rate decay
- ResNet50, ResNet101, every combination (width, batch size) of
    - Width: $8, 16$
    - Batch size: $64, 256$, without learning rate decay
- VGG11, every combination (width, batch size) of
    - Width: $16, 64$
    - Batch size: $64, 256$, without learning rate decay
- DenseNet40, every combination (width, batch size) of
    - Width: $5, 12, 40$
    - Batch size: $64, 256$, without learning rate decay

For models in Figure 4, more than 5 classifiers were trained. The classifiers differ in weight initialization and vary due to the randomized batches used during training.

- ResNet18 on CIFAR-10, width 16 and batch size 64 without learning rate decay (20 classifiers)

The models below are trained with learning rate 0.05, momentum 0.9 and weight decay 5e-4 with cosine annealing.

- MobileNet on MNIST, batch size 128 (10 classifiers)
- ResNet18 on FMNIST, width 48 and batch size 128 (10 classifiers)
- ResNet18 on KMNIST, every combination of widths and batch sizes below (8 classifiers each)
    - Width: $48, 64$
    - Batch size: $32, 64, 128$

## C.2 Majority vote and tie-free

For an ensemble with $N$ classifiers, we generated $N$ uniformly-distributed random numbers $e_1, ..., e_N \in [0, 0.0001]$. Then used $(\frac{1}{N} + e_1, ... \frac{1}{N} + e_N)$ after normalization as weights for each classifier. This guarantees the ensemble to be tie-free.

