# OpenReview forum: "How many classifiers do we need?"
_NeurIPS.cc/2024/Conference — NeurIPS 2024 poster_

### Official Review · Reviewer_2Qk3 · 2024-06-20

**Soundness:** 4
**Presentation:** 3
**Contribution:** 3
**Rating:** 7
**Confidence:** 3

**Summary:**

The authors address the setting of ensembling, where the predictions of multiple models are combined to improve accuracy and form more robust conclusions. The authors define a quantity η (polarity among agents within a dataset), and show both empirically and theoretically that this quantity is nearly constant regardless of hyperparameters or architectures of classifiers. The authors present a tight upper bound for the error of majority vote under restricted entropy conditions. This bound indicates that the disagreement is linearly correlated with the target, and the slope is linear in polarity, η. Finally, the authors prove asymptotic behavior of disagreement in terms of the number of agents, which can help predicting the performance for a larger number of agents from that of a smaller number.

**Strengths:**

1. Section 2 is, by in large, excellent, and does a very good job of summarizing the existing work in this area.
2. The concept of polarity is clearly defined and well supported via the helpful example in Fig. 1.
3. The tighter bounds in Sec. 4 are well-described and scoped.

**Weaknesses:**

1. The claim that the authors have shown empirically that η is nearly constant regardless of hparams or architectures is overbroad for the provided evidence; the authors should consider scoping their empirical claims more modestly, in keeping with the small scale experiments in this paper.
2. The phrase "a distribution over a parametric family hθ , e.g., a distribution of classifiers that is trained from a neural network or a random forest", could be misinterpreted as a claim that random forests are parametric models
3. Equation (6) appears to have a formatting error, there's a long underscore under one half of the equation
4. Conjecture 1 uses the term *interpolating* neural networks; however, this term is not defined in Sec. 2 and is overloaded in the literature. Please provide a definition in Sec. 2 or a relevant citation.
5. "Since the denominator ... is invariant to the number of classifiers and the numerator resembles the disagreement between classifiers, δ is expected to follow a similar pattern": this point could use more clarification

**Questions:**

QUESTIONS

* How do the authors define *interpolating* neural networks in this work?
* Can the authors please expand on the point raised in W5?

SUMMARY OF THE REVIEW

This paper presents significant new theoretical contributions in its area, which is important and of interest to the larger community, but has a few overclaims and oversights which currently limit it. As the paper stands, I will give a weak accept, but if the weaknesses I list are addressed, I will raise my score.

**Limitations:**

The authors have adequately addressed the limitations and potential negative societal impact of their work.

---

> ### Author Rebuttal · Authors · 2024-08-07
>
> Thank you for reviewing this paper and we really appreciate your comments and questions.
>
> [Weakness 1] Agreed. That was the reason why we narrowed down the scope to interpolating models in Conjecture 1 although Theorem 1 holds for any ensemble. We think it would be an interesting direction to see how $\eta$ differs in non-interpolating models in different fields (e.g. language models).
>
> [Weakness 2] Thank you for catching this. We will fix this in the final version.
>
> [Weakness 3] The long underscore is due to the footnote below the formula.
>
> [Weakness 4, Question 1] Please see the general response: "interpolator" is a model that is trained to perfectly fit the train data. In Conjecture 1, we wanted to draw a statement analogous to so-called 'neural scaling laws', which states that the out-of-sample performance of an interpolating neural network scales with the width of the neural net. We will add a paragraph about this on the final version.
>
> [Weakness 5, Question 2] In Theorem 5, we prove that the disagreement for $N$ classifiers has a representation as $(1-1/N)(D_\infty + N^{-1/2}Z_N)$, and through Donsker's invariance principle, it can be approximated through the disagreement of a smaller number of classifiers (say $M$) as $(1-1/N) / (1-1/M) * (\text{Disg. of }M\text{ classifiers})$. The justification for this argument is that the disagreement is a $V$-statistic, and so its bias can be accounted for. The same is true for the numerator of $\delta$ (the denominator is unbiased): it is also a $V$-statistic, and should behave similarly to the disagreement, as its form is the same with only one less label. We will add this clarification in the final version.

---

> ### Comment · Reviewer_2Qk3 · 2024-08-08
>
> I would like to thank the authors for their detailed response. To acknowledge that my concerns have been addressed, I will update my score.

---

> > ### Author Response · Authors · 2024-08-13
> >
> > We are glad your concerns were addressed. Please let us know if you have any additional questions.

---

### Official Review · Reviewer_CG1s · 2024-06-27

**Soundness:** 3
**Presentation:** 2
**Contribution:** 3
**Rating:** 6
**Confidence:** 4

**Summary:**

This paper analyzes the majority vote error of an ensemble of classifiers. A new quantity called polarity is introduced. The polarity of neural networks is analyzed empirically and theoretically, and stronger bounds on the majority vote error are derived based on the polarity. Finally, the previously derived bounds are used to predict the majority vote error rate of a large ensemble from that of a smaller ensemble.

**Strengths:**

As far as I am aware, the notion of an ensemble's polarity around which the paper is developed is a novel concept. The derived bounds based on polarity are stronger than previously known bounds and even subsume some of them, for example those based on the notion of competent ensembles.

**Weaknesses:**

The conjecture of the neural polarity is somewhat lacking in evidence given the generality of the statement. While Figure 1 provides some empirical evidence for a variety of models and hyperparameters, only the relatively simple CIFAR-10 dataset is considered. It would be interesting to see if the neural polarity law still holds for more challenging tasks, where the performance of each individual classifier would be lower.

I also feel that some parts of the paper could be improved for clarity. In particular, in Section 5 it is somewhat unclear how all the results fit together, which can detract from their potential significance. See some of the questions below.

**Questions:**

1. The description of Theorem 1 as a lower bound in line 127 is somewhat misleading. As mentioned later in the paper, models can be polarized for $\eta$ lower than the derived bound. I wonder if the wording here can be improved.

2. I do not quite see the connection to neural collapse in line 159. Neural collapse describes the phenomenon that representations of examples from the same class collapse to a single point and the class means form a simplex ETF. Still, it does not state how the probabilities of any given example are distributed among the labels.

3. In theorem 5, what is $\sigma^2$? And what is the significance of the scaled random walk converging to Brownian motion? As far as I can tell, the convergence to Brownian motion is a restatement of a standard result from stochastic calculus and is not used anywhere else in the paper.

4. In the definition of $\mathcal{L}(h)$ in line 215, it appears that the symbol $h$ is overloaded.

**Limitations:**

Limitations are addressed.

---

> ### Author Rebuttal · Authors · 2024-08-07
>
> Thank you for reviewing this paper; we really appreciate your comments and questions.
>
> [Weakness 1] We agree that further empirical evidence would strengthen our claim. It is difficult to obtain an ensemble of well-trained interpolating models in practice, so we are limited in our selection of models and datasets. For now, we provide further evidence of the universality of $\eta<4/3$ on two additional datasets in the figure document: KMNIST and Fashion-MNIST. Please refer to Figure A on the figure document attached to the rebuttal for all reviewers above.
>
> On KMNIST, which is an easier task than CIFAR-10, ResNet18 with various width (48, 64) were trained with various batch size (32,64,128) to interpolate the train data.
> On Fashion-MNIST, ResNet18 with width 48 was trained with batch size 128 to interpolate the train data.
>
> [Weakness 2, Question 3] The main role of Theorem 5 is to justify our approximation of the entire disagreement curve by neglecting $N^{-1/2}Z_N$. If we used a ordinary central limit theorem, we could only justify approximations of $D_n$ for large $n$. Appealing to Donsker's invariance principle, we can justify our procedure of extrapolating the disagreement as $D_n = D_3 * (1-1/n)/(1-1/3)$.
>
> In this case, $\sigma^2$ is equal to $Var_{h\sim\rho}(L(h))$. We will update this in the final version along with the typo on the $U_N$ Hoeffding decomposition in Line 214 where $E_{\rho_N^2}$ should be $E_{\rho^2}$ instead.
>
> [Question 1] We rephrased the theorem and conjecture and posted as a rebuttal for all reviewers above. We hope this clarifies the statement.
>
> [Question 2] Perhaps it is only at an intuitive level, but we see neural collapse and reduced entropy on the probabilities as symptoms of the same general phenomenon. If the representations within each class are reduced to a single point, then a classifier ensemble is going to become increasingly confident within each class, as there is less variability in the outputs.
>
> [Question 3] Answered with Weakness 2 above.
>
> [Question 4] Thank you for pointing this out. We will update this to $\mathcal{L}(h)=\mathbb{E}\_{h'\sim\rho}\mathbb{P}\_{\mathcal{D}}(h(X)\neq h'(X))-\mathbb{E}\_{(h',h'')\sim\rho^2}\mathbb{P}\_{\mathcal{D}}(h'(X)\neq h''(X))$ in the final version.

---

> > ### Comment · Reviewer_CG1s · 2024-08-08
> >
> > Thank you for the additional results and clarifications. I would be glad to increase my score.

---

> > > ### Author Response · Authors · 2024-08-13
> > >
> > > We are glad the additional results and clarification were helpful. Please let us know if you have any additional questions or need further clarification.

---

### Official Review · Reviewer_ueod · 2024-07-09

**Soundness:** 3
**Presentation:** 3
**Contribution:** 3
**Rating:** 7
**Confidence:** 2

**Summary:**

This paper focuses on quantifying the impact of number of classifiers on the error rate of majority vote decision strategy for ensemble classifiers. They define the notion of “polarity” of an ensemble and use this notion to characterize the relationship between majority vote error rate and disagreement between classifiers in any relevant ensemble. Empirical analysis over CIFAR and MNIST datasets shows that the paper’s method provides a tighter bound for majority error vote than other prior methods.

**Strengths:**

1. Overall, the paper is well-written, concise, and would be valuable to the wide audience working on predictive optimization and ensemble models.
2. The use of the notion of polarity is novel and seems to capture two relevant properties simultaneously: (a) whether a large fraction of classifiers are making an incorrect prediction, and (b) whether the predictions of large groups of classifiers differ from each other. This characterization specially seems to assist in handling the presence of multiple classes as well. The conjectured upper bound on polarity is also independently interesting.
3. I liked the focus on entropy-restricted ensembles. The focus on ensembles with quantifiable “disagreement properties“ (as in Thms 3, 4) allows for stronger bounds on majority vote error rate, although it does seem difficult to then derive bounds for these “disagreement properties” (like the parameter $\delta$). Nevertheless, the approach of bounding majority vote error rate using disagreement notions is intuitively appealing and, as the paper shows, it does provide reasonable theoretical bounds.

**Weaknesses:**

1. Some of the terms used in the paper could use additional descriptions. For example, I don’t see the description of “interpolation” or “interpolating models” anywhere. Considering that it is important in interpreting Figure 1 and also determining when the conjectured polarity upper bound is likely to hold, I would suggest spending a paragraph on prior work around interpolation.

2. Related to the above point, some more details around the Remark in Lines 139-145 would be helpful. It looks like for the non-constant term in the maximum of Thm 1, the numerator and denominator are empirical approximations of the numerator and denominator of $\eta$. Given that, could these converge to $\eta$ itself as $m \rightarrow \infty$? And if so, isn’t it possible for the Conjecture 1 to be violated as even values of $\eta > 4/3$ could satisfy the condition in Theorem 1? Let me know if I am misinterpreting something here.
3. More details of empirical analysis would be good to include in the main body. For instance, details about number of classifiers in each ensemble and whether the classifiers only differed in starting points (or other parameters as well) would be good to know while reading the figures themselves

**Questions:**

1. On the point of polarity, it would be good to get more details around the discussion in the Remark on Lines 139-145. The specific questions I have on this are noted in the point above.
2. Regarding my point on empirical analysis, did the trained classifier ensemble consist of classifiers with just different starting points or were other training details/parameters changed?
3. Minor point, but $\delta$ notation is used in both Theorem 1 (for probability) and Theorems 3, 4 (for disagreement). And it looks like this notation is used in different contexts in these different results. Might be good to use different notations at different places if they are not related.

**Limitations:**

Some limitations discussed in the conclusion section.

---

> ### Author Rebuttal · Authors · 2024-08-07
>
> Thank you for your positive appraisal of our work!
>
> [Weakness 1] Please see the general response. Interpolators are models that are trained to perfectly fit the train data, and appear prominently in the double descent literature, as well as prior explorations into the benefits of ensembling.
>
> [Weakness 2, Question 1] Thank you for pointing this out. It is true that the empirical approximation converges to $\eta$ itself. Here the bound becomes trivial with large $m$, as $\eta$ is clearly bounded by the maximum of $4/3$ and itself.
>
> We rephrased the theorem and conjecture and posted as a rebuttal for all reviewers above. We hope this clarifies the statement. Our conjecture follows from the observation that $P/S$ is less than $4/3$ (and hence $\max\{4/3, P/S\}$) for interpolators. To prove this in the theory is a fascinating open problem for us.
>
> [Weakness 3, Question 2] We will add more implementation details to Appendix C in the final version. The constituent classifiers differ in weight initialization and vary due to the randomized batches used during training. We see that the number of classifiers we used (mostly 5) is specified in Appendix C.1, but please kindly let us know if you feel we missed something.
>
> [Question 3] Thank you for pointing this out. We will correct this in the final version.

---

> > ### Comment · Reviewer_ueod · 2024-08-10
> >
> > Thanks to the authors for the clarifications. I will keep my score as it is but I would suggest moving some of the implementation details from the appendix to the main body if possible.

---

> > > ### Author Response · Authors · 2024-08-13
> > >
> > > We are glad that we could address your concerns and questions regarding our work. We will move some key implementation details to the main body in the final version.

---

### Official Review · Reviewer_UfPx · 2024-07-13

**Soundness:** 3
**Presentation:** 3
**Contribution:** 2
**Rating:** 7
**Confidence:** 3

**Summary:**

The authors introduce polarity $\eta$ to characterize an ensemble of classifiers. Polarity is the probability that over half the models in an ensemble make an incorrect prediction divided by the expected square fraction of models that make an incorrect prediction. This quantity is bound by a concentration inequality, and the authors empirically demonstrate that this value tends to hover around 4/3. The authors then bound the ensemble error probability via a relation that involves polarity, and refine this bound based on further assumptions on the behavior of the ensemble. Finally, the authors show how to extrapolate small ensemble statistics to estimate error rates of larger ensembles.

**Strengths:**

The work is well motivated and clearly presented. The authors show that the theorems they derive can be used to estimate the performance of large ensembles (although further validation of this would strengthen the paper, see weaknesses).

**Weaknesses:**

It seems that Theorem 1 would hold if we replace 4/3 with any quantity that is larger than 4/3 as well, so theoretically it is unclear where this quantity really arises from and what prevents this quantity from being made smaller. Perhaps the authors can sketch out some intuition on this.

Some additional experimental validation could help strengthen paper, since some of the results are empirical. For example, the experiments shown in Figure 3 could be extended to the other models discussed in the paper, and ideally should be performed multiple times to obtain a good idea of how reliable the estimates will be. Furthermore, some statistics could be reported (e.g. mean and variances for the distributions in Figure 1). However this is primarily theoretical work, so this is not a major concern.

**Questions:**

Figure 1 appears to show a positive correlation between error rate and polarity. I imagine this is driven by the numerator term in polarity. Could the authors comment on this, and whether they would expect the 4/3 rule to hold in a small or large error regime (e.g. $\epsilon$ or $0.5-\epsilon$?).

For the experimental results shown in Figure 3, is it possible to plot some of the other bounds described in Section 2.2 for reference?

The ``#’’ sign shows up in an equation on line 131, which was not defined. Is this an error?

**Limitations:**

Yes.

---

> ### Author Rebuttal · Authors · 2024-08-07
>
> Thank you for your positive review of our paper; we greatly appreciate your comments and questions.
>
> [Weakness 1] Please see the general response: we hope that the newly presented form of Theorem 1 addresses your concerns. In the original wording, we state that an $\eta$ greater than the stated lower bound is guaranteed. This is true, but one should pick the best possible $\eta$ for the task, which is what the newly worded theorem now states. $4/3$ in the theorem mainly arises from Lemma 1, equation (9). The notable point is that the value $4/3$ aligns well with the empirical result as shown in Figure 1.
>
> [Weakness 2] Thank you for your comments. Please refer to Figure C on the figure document attached to the rebuttal for all reviewers above. On KMNIST, which is an easier task than CIFAR-10, ResNet18 with various width (48, 64) were trained with batch size (32,64,128) to interpolate the train data. On Fashion-MNIST, ResNet18 with width 48 was trained with batch size 128 to interpolate the train data.
>
> [Question 1] Lemma 2 states that the majority vote error rate is smaller than $P(W_\rho > 1/2)$, the numerator term in the definition of polarity. If the error W_\rho(X) is smaller than 0.5 for all datapoints in the test set, the equality holds and the majority vote error rate $= P(W_\rho > 1/2) = 0$. This ensemble is $0$-polarized and thus $4/3$-polarized. We may have misunderstood your question. Could you explain a bit more about what you meant by 'small or large error regime (e.g. $\epsilon$ or $0.5-\epsilon$)'?
>
> [Question 2] We avoided overlaying those bounds (described in Section 2.2) as they are so far above the current set of axes that one cannot deduce the accuracy of our novel bounds. Please refer to Figure B in the figure document attached to the rebuttal for all reviewers above
>
> [Question 3] We used # as an indicator function 'f(A)=1 if A is true, otherwise 0'. For clarity, we will change this in the final version to a more common notation $1_A$.

---

> > ### Comment · Reviewer_UfPx · 2024-08-11
> >
> > Thank you for your response. I remain unsure whether 4/3 is an intrinsic property of polarity of whether it is an artifact of the proof - I would need to spend more time with the paper, which unfortunately we are short on. I am satisfied with the new figures the authors have provided to address [Weakness 2] and I have updated my score to reflect this.
> >
> > On question 1: If you look at Figure 1 it seems as though there is a positive correlation. I was asking what you would expect to happen if the x-axis is expanded to 0-0.5 and populated with additional points in those unseen regimes.

---

> ### Author Response · Authors · 2024-08-14
>
> Our current theory doesn’t suggest a _general_ positive correlation. This is reflected in our experiments: when comparing the CIFAR10 and CIFAR10-1 examples in Figure A ([figure_document](https://openreview.net/attachment?id=RvSCTfKipG&name=pdf)), the polarity of the interpolators hasn’t changed significantly, even though the majority vote error rate has increased by ~0.1 on average.
>
> However, we do not reject the possibility that a correlation might exist on a case-by-case basis. Our listed examples, Example 1 and Example 2 on page 4, highlight two extreme cases where this correlation is present. We are interested in conducting further investigations in future work.
>
> Regarding polarity, as both Figure 1 and Figure A ([figure_document](https://openreview.net/attachment?id=RvSCTfKipG&name=pdf)) suggest, and based on our efforts to make Lemma 1 reasonably sharp, we conjecture that 4/3 is not purely an artifact of the proof.

---

### Author Rebuttal · Authors · 2024-08-07

As there are overlapping comments regarding Theorem 1 and Conjecture 1, we provide clarification and a rephrased version of the theorem and conjecture here. Firstly, for Theorem 1, we recognize that the current wording may be misleading, so we reword the theorem in the following way for clarity:

Theorem 1: Letting $$\eta = \max\left\\{ \frac{4}{3},\left(\frac{\sqrt{\frac{3}{8m}\log\frac{1}{\delta}}+\sqrt{\frac{3}{8m}\log\frac{1}{\delta}+4SP}}{2S}\right)^{2}\right\\}$$, the ensemble $\rho$ is $\eta$-polarized with probability at least $1-\delta$.

No modifications to the proof are necessary, as this is simply a restatement of the finding obtained at the end of the proof.

Definition of Interpolator, Interpolating: We define a neural net (or any other type of model) as an interpolator and say it is interpolating if it is trained to exactly fit the training data, i.e. $L_{train} = 0$. This term was recently popularized in Belkin et al.'s paper, "Reconciling Modern Machine-Learning Practice and the Classical Bias–Variance Trade-Off", which we now cite.

Conjecture 1: An ensemble $\rho$ comprised of independently trained high-performing interpolating neural networks is $\eta$-polarized for $\eta \leq 4/3$.

As some reviewers have pointed out, the previous wording suggested that interpolating neural network ensembles were $\eta$-polarized for $\eta$ precisely equal to $4/3$ and no lower. To clarify, we are asserting that $4/3$ suffices, as evidenced by Figure 1, which highlights that estimates for polarity (that is, $P/S$) are consistently below $4/3$ for interpolating models. We also wish to add that the "high-performing" moniker should be necessary here to ensure that $P$ is not too large in general.

---

### Decision · Program_Chairs · 2024-09-25

**Decision:**

Accept (poster)

**Comment:**

The reviewers agreed that this paper addresses an important topic, is clearly written, and has strong results. Some reviewers had concerns about the variety of tasks, models, and datasets in the experimental validations, but thought the positives outweighed this concern. Given the unanimous recommendation for acceptance from the four reviewers, I recommend accepting this paper.